# The Effect of Angiogenesis-Based Scaffold of MesoporousBioactive Glass Nanofiber on Osteogenesis

**DOI:** 10.3390/ijms232012670

**Published:** 2022-10-21

**Authors:** Weijia Zheng, Zhenzu Bai, Shan Huang, Kai Jiang, Long Liu, Xiaoyan Wang

**Affiliations:** Department of Biology and Chemistry, College of Science, National University of Defense Technology, Changsha 410073, China

**Keywords:** scaffold, coaxial electrospinning, osteogenesis, angiogenesis, bone defect repair

## Abstract

There is still an urgent need for more efficient biological scaffolds to promote the healing of bone defects. Vessels can accelerate bone growth and regeneration by transporting nutrients, which is an excellent method to jointly increase osteogenesis and angiogenesis in bone regeneration. Therefore, we aimed to prepare a composite scaffold that could promote osteogenesis with angiogenesis to enhance bone defect repair. Here, we report that scaffolds were prepared by coaxial electrospinning with mesoporous bioactive glass modified with amino (MBG-NH_2_) adsorbing insulin-like growth factor-1 (IGF-1) as the core and silk fibroin (SF) adsorbing vascular endothelial growth factor (VEGF) as the shell. These scaffolds were named MBG-NH_2_/IGF@SF/VEGF and might be used as repair materials to promote bone defect repair. Interestingly, we found that the MBG-NH_2_/IGF@SF/VEGF scaffolds had nano-scale morphology and high porosity, as well as enough mechanical strength to support the tissue. Moreover, MBG-NH_2_ could sustain the release of IGF-1 to achieve long-term repair. Additionally, the MBG-NH_2_/IGF@SF/VEGF scaffolds could significantly promote the mRNA expression levels of osteogenic marker genes and the protein expression levels of Bmp2 and Runx2 in bone marrow mesenchymal stem cells (BMSCs). Meanwhile, the MBG-NH_2_/IGF@SF/VEGF scaffolds promoted osteogenesis by simulating Runx2 transcription activity through the phosphorylated Erk1/2-activated pathway. Intriguingly, the MBG-NH_2_/IGF@SF/VEGF scaffolds could also significantly promote the mRNA expression level of angiogenesis marker genes and the protein expression level of CD31. Furthermore, RNA sequencing verified that the MBG-NH_2_/IGF@SF/VEGF scaffolds had excellent performance in promoting bone defect repair and angiogenesis. Consistent with these observations, we found that the MBG-NH_2_/IGF@SF/VEGF scaffolds demonstrated a good repair effect on a critical skull defect in mice *in vivo*, which not only promoted the formation of blood vessels in the haversian canal but also accelerated the bone repair process. We concluded that these MBG-NH_2_/IGF@SF/VEGF scaffolds could promote bone defect repair under accelerating angiogenesis. Our finding provides a new potential biomaterial for bone tissue engineering.

## 1. Introduction

Bones are important for the human body. However, a bone defect is a common but intractable disease in clinics that usually results from trauma, resection and genetic disorders [1]. Currently, critical bone defects are still great challenges in the clinic. Although there is no doubt that autologous bone transplants comprise the best method for the clinical treatment of critical bone defects, their sources are limited and it is easy to induce complications and secondary injuries. Furthermore, allogeneic or xenogeneic bone has risks of viral disease transmission and immune rejection, and there is often a lack of donors. In order to solve this problem, research on bone substitute materials with excellent biochemical properties and controllable costs has become a hot spot.

The key point of bone tissue engineering is the construction of scaffolds. The main parts of scaffolds can be summarized as follows: scaffold materials, cells, and growth factors. It is conducive to prepare excellent osteogenic scaffolds through a combination of a superior materials and effective growth factors. Common materials include inorganic materials such as hydroxyapatite (HA) [2,3,4], titanium alloy [5,6], and tricalcium phosphate (TCP) [7,8] and organic materials such as polylactic acid (PLA) [9], chitosan (CS) [10,11], collagen [12], and silk fibroin (SF) [13,14,15]. In early studies, these materials were often used alone, but in recent studies, two to three or even more materials have been combined. Furthermore, biological scaffolds have prepared by various methods including layer-by-layer (LBL) self-assembly [16,17], 3D printing [4,6,7,9], coating [7,16] and electrospinning [13,18].

An emergent material, mesoporous bioactive glass (MBG) has excellent bioactivity, biocompatibility and bone conductivity for repairing bone defects. The introduction of a pore-enlarging agent and template during preparation provides MBG with a regular porosity and a large specific surface area [19,20]. MBG can adsorb a large number of drugs, proteins or growth factors due a porosity range from 2 to 50 nm, and it can persistently release them to accelerate osteoblast differentiation and osteogenesis [21,22,23,24]. Its large specific surface area is also conducive to cell adhesion and material exchange, promoting cell proliferation [25] and the mineralization of matrix nodules and accelerating the process of bone repair [26]. Moreover, it is reported that MBG modified with amino (MBG-NH_2_) has better biocompatibility, factor-binding capacity, and bone-promoting ability qualities than MBG alone [23]. The combination of MBG and SF can promote type I collagen synthesis and osteogenesis [27].

Bone mainly consists of complex of type I collagen and nano-HA with cells growing on it [28]. An extracellular matrix (ECM) has a large content of collagen. An ECM is usually a high-porosity network structure consisting of nanofibers that can regulate the cellular physiological process. It has been found that the smaller the diameter of nanofibers, the higher the degree of the differentiation of cells growing on them [29]. Nanofiber scaffolds can be used as abiotic substitutes for ECMs to offer better surroundings for cell proliferation, differentiation, mineralization, as well as the intercellular signal pathway, *in vitro*.

A high-voltage direct current (DC) power supply, digital syringe pump, and collector are the main components of electrospinning equipment. The working principle of electrospinning is that a charged polymer fluid produces a jet flow to form fiber under the driving force of a high-voltage electrostatic field [30]. Under the driving force of this electric field, the droplets squeezed out of the needle tip form a spike pointing to the collector and further extending to the collector in the form of a Taylor cone. During the extension process, the diameter is continuously reduced until nanofibers are finally prepared [31]. The biological scaffolds prepared by electrospinning are composed of nanofiber stacks with basically even diameters, and they mostly have uniform structures and a high porosity. During spinning, high-molecular-weight materials with stable properties and good biocompatibility can be used as raw materials.

Moreover, osteogenic growth factors or inorganic salts can be added. For instance, IGF-1 [32,33,34], which has been identified to promote osteogenesis, can be used to promote the proliferation, differentiation and mineralization of cells cultured on scaffolds. It has been found that IGF-1 can activate the mTOR signaling pathway during osteogenic differentiation. Furthermore, IGF-1 can increase the mRNA expression levels of osteogenesis maker genes [35] and raise the expression of bone morphogenetic protein 2 (*Bmp2*) and osteocalcin (*Ocn*) through the mitogen-activated protein kinase (MAPK) signaling pathway [36,37,38]. Therefore, IGF-1 has good application prospects and is worth studying in bone tissue engineering.

The process of neovascularization is also indispensable in osteogenesis. The reconstruction of the vascular network at a bone defect will offer a good micro-environment for the related physiological process of osteogenesis. Previous studies have shown that the vascular distribution in a bone defect is a critical factor in the repair process, and VEGF is an important factor for the reconstruction of a bone defect vascular network [39]. VEGF is highly specific and can promote increases in vascular permeability, ECM degeneration, the migration and proliferation of vascular endothelial cells, and angiogenesis [40], so it has been used to construct bone repair scaffolds with excellent osteogenic properties [41,42]. Therefore, it is also good to add VEGF to scaffolds to provide the function of promoting angiogenesis during preparation, which could greatly improve the osteogenic performance of scaffolds.

To sum up, in this study, biological scaffolds were prepared with the electrospinning method, in which MBG-NH_2_ adsorbed with IGF-1 was used as the core and SF adsorbed with VEGF was used as the shell. The biological scaffolds were used for the culture of bone marrow mesenchymal stem cells (BMSCs) and human umbilical vein endothelial cells (HUVECs) to verify their osteogenic and angiogenic effects, respectively, and the promoting effect of angiogenesis on osteogenesis was explored by sequencing after the co-culture of these two cells. Furthermore, we also evaluated the in vivo bone repair performance of the scaffolds (Figure 1).

## 2. Results

### 2.1. Characterization and Bioactivity of Scaffolds

#### 2.1.1. Characterization of Scaffolds

SEM was used to determine the morphology and microstructure of the MBG-NH_2_, MBG-NH_2_/IGF and MBG-NH_2_/IGF@SF/VEGF scaffolds. It was shown that all these scaffolds contained fibers of uniform size and nanometer diameter. The fibers were stacked to form a three-dimensional network with a high porosity that could benefit cell adhesion and growth. The surfaces of the MBG-NH_2_ and MBG-NH_2_/IGF scaffolds were rough, and there were small parts of raised MBG particles. In particular, the MBG-NH_2_/IGF@SF/VEGF scaffolds had a relatively smooth surface (Figure 1A) because they had an outer shell of SF covering the bioactive glass inside.

In order to further confirm the core–shell structure of the coaxial fiber, we labeled the core and shell with green and red fluorescence, respectively, when preparing the coaxial fiber. Under CLSM, we could observe the co-localization of the green core and the red shell, and it could be observed in the merged image that the overlap between the core and the shell presented as yellow, which could be well-discerned in both plan view and perspective view (Figure 1B). This proved that the coaxial fiber had a core–shell structure.

Furthermore, the FTIR spectrometry of the scaffolds indicated that the PCL, MBG-NH_2_, MBG-NH_2_/IGF, and MBG-NH_2_/IGF@SF/VEGF scaffolds showed similar characteristic peaks (Figure 1C). There were peaks at 1500 cm^−1^ and 1600 cm^−1^ of the MBG-NH_2_/IGF@SF/VEGF scaffolds that were different from those of MBG-NH_2_ and MBG-NH_2_/IGF (Figure 1C). These two peaks corresponded to the amide bond in the β-pleated sheet structure of SF, which were type II and type I, respectively. Additionally, the peak at 3290 cm^−1^ implied the existence of an amide group (Figure 1C). These characteristic peaks were all higher than the peaks of SF. This result suggests that the introduction of SF and growth factors did not significantly change the chemical structure of scaffolds, as they were adsorbed onto the scaffolds through physical force.

Since the scaffold should play a supporting structure in the process of osteogenesis, to assess the mechanical strength of the scaffolds, their mechanical properties were tested with a tensile test. It was shown that the elastic modulus of the MBG-NH_2_/IGF@SF/VEGF scaffolds was significantly better than that of the other two scaffolds (Figure 1D; compare column 3 with columns 2 and 1), which suggested that the introduction of IGF-1 and SF could both enhance the toughness of the scaffolds and provide better support for the bone defect repair.

#### 2.1.2. Bioactivity of Scaffolds

To study the release sustainability of MBG-NH_2_, we tested the release ability of IGF-1. The results showed that MBG-NH_2_ was slowly releasing IGF-1 (Figure 1E), which proved that MBG-NH_2_ had a strong loading capacity and that the composite scaffolds could sustain the release growth factors for bone defects.

We detected the deposition particles formed on the scaffolds with SEM to determine the calcium deposition on the scaffolds in simulated body fluid (SBF). It was shown that there was almost no calcium deposition on the MBG-NH_2_ scaffolds, though there were deposition particles on the other two scaffolds, and the amount of calcium deposition on the MBG-NH_2_/IGF@SF/VEGF scaffolds was significantly greater than that on the MBG-NH_2_/IGF scaffolds (Figure 1F). These results proved that the MBG-NH_2_/IGF and MBG-NH_2_/IGF@SF/VEGF scaffolds had the highest calcium deposition abilities *in vitro*, and that of the MBG-NH_2_/IGF@SF/VEGF scaffolds was the best.

### 2.2. BMSCs Proliferation, Differentiation and Mineralization Cultured on These Scaffolds

In order to study the effect of the scaffolds on BMSCs, we evaluated the proliferation, differentiation and mineralization in BMSCs cultured on these scaffolds. Firstly, we evaluated cell proliferation by cell activity. It was shown that the cell activity of the BMSCs cultured on the MBG-NH_2_/IGF@SF/VEGF scaffolds rapidly increased within 7 days (Figure 2A; compare the black columns with the twill shading column and the white column), indicating that the MBG-NH_2_/IGF@SF/VEGF scaffolds had a good effect on promoting proliferation. Since ALP activity increases during early osteoblast differentiation [43], ALP activity can be used to evaluate cell differentiation. In the process of osteoblast differentiation, the ALP activity of the BMSCs cultured on these scaffolds reached a high level on day 3 and then decreased (Figure 2B). Additionally, the ALP activities of the BMSCs cultured on the MBG-NH_2_/IGF@SF/VEGF scaffolds on day 6 were higher than those of the other two scaffolds (Figure 2B; compare the black columns with the twill shading column and the white column), indicating that the MBG-NH_2_/IGF@SF/VEGF scaffolds were better at promoting osteoblast differentiation. The amount and area of alizarin red-stained mineralized nodules demonstrated the degree of mineralization of the BMSCs cultured on these scaffolds. The results showed that compared with those on the MBG-NH_2_ scaffolds, the BMSCs cultured on the other two scaffolds had larger mineralized nodules; the mineralized nodules of the BMSCs cultured on the MBG-NH_2_/IGF@SF/VEGF scaffolds were also of a higher amount and were more densely packed (Figure 2C), which indicated that both the MBG-NH_2_ scaffolds combined with IGF-1 and the coaxial structure fiber scaffolds combined with IGF-1 and VEGF had a better ability to promote the mineralization of the BMSCs and that the performance of the MBG-NH_2_/IGF@SF/VEGF scaffolds was better than that of the MBG-NH_2_/IGF scaffolds.

### 2.3. The Effect of Scaffolds on Osteogenesis In Vitro

#### 2.3.1. Expression of Osteogenic Genes and Runx2 Transcriptional Activity

In order to determine the effect of scaffolds on osteogenesis, we tested the mRNA expression levels of osteoblastic-specific marker genes in the BMSCs cultured on these scaffolds. It was shown that compared with those of the BMSCs cultured on the MBG-NH_2_ scaffolds, the mRNA expression levels of *Bmp2* (Figure 3A), *Opn* (Figure 3B), *Osterix* (Figure 3C) and *Runx2* (Figure 3D) in the BMSCs cultured on the other two scaffolds were significantly increased (Figure 3A–D; compare columns 3 and 2 with column 1) and the mRNA expression levels in the BMSCs cultured on the MBG-NH_2_/IGF@SF/VEGF scaffolds were obviously the highest (Figure 3A–D, column 3). These results suggested that the MBG-NH_2_/IGF and MBG-NH_2_/IGF@SF/VEGF scaffolds increased the mRNA expression levels of osteoblast-specific marker genes in the BMSCs at the transcriptional level and that the MBG-NH2/IGF@SF/VEGF scaffolds were superior to the MBG-NH2/IGF scaffolds.

Runx2 is an important transcription factor for osteoblast differentiation [44], so the ability of scaffolds to promote bone healing can be evaluated by whether the scaffolds can affect the transcriptional activity of Runx2 in BMSCs. The results of the luciferase reporter gene test showed that the transcriptional activity of Runx2 in the BMSCs cultured on the MBG-NH_2_/IGF@SF/VEGF scaffolds was significantly higher than that on the MBG-NH_2_ scaffolds (Figure 3E; compare column 3 with column 1). Therefore, the MBG-NH_2_/IGF and MBG-NH_2_/IGF@SF/VEGF scaffolds increased the expression of osteoblast-specific marker genes and Runx2 transcriptional activity in the BMSCs, and the MBG-NH_2_/IGF@SF/VEGF scaffolds showed a superior performance to the MBG-NH_2_/IGF scaffolds.

#### 2.3.2. Expression of Osteogenic Proteins

While Runx2 is activated by p-Erk1/2 signaling, the ser^301^ and ser^319^ residues are phosphorylated [45]. Furthermore, BMSCs can be induced to differentiate into osteoblasts by phosphorylated mammalian target of rapamycin (p-mTOR) [38]. To investigate the mechanism of scaffolds promoting osteogenesis at the protein level, we evaluated the content of osteogenic-specific marker proteins in the BMSCs cultured on scaffolds. It was shown that compared with the MBG-NH_2_ scaffolds, the expression of p-Erk1/2 in the BMSCs cultured on the MBG-NH_2_/IGF and MBG-NH_2_/IGF@SF/VEGF scaffolds was higher (Figure 3F; compare lines 2 and 3 with line 1) and that the expression on the former was the highest (Figure 3F, line 2). Similarly, compared with the MBG-NH_2_ scaffolds, the p-mTOR expression of the BMSCs cultured on the MBG-NH_2_/IGF and MBG-NH_2_/IGF@SF/VEGF scaffolds were higher (Figure 3F; compare lines 2 and 3 with line 1), and that of the MBG-NH_2_/IGF scaffolds was superior to that of MBG-NH_2_/IGF@SF/VEGF (Figure 3F; compare line 2 with line 3). These results demonstrate that IGF-1 in the MBG-NH_2_/IGF and MBG-NH_2_/IGF@SF/VEGF scaffolds activates the p-mTOR-regulated pathway to regulate osteoblast differentiation in BMSCs and that the MBG-NH_2_/IGF and MBG-NH_2_/IGF@SF/VEGF scaffolds could induce the p-Erk1/2-activated Runx2 pathway. The immunofluorescence results showed that the fluorescence signal intensity and fluorescence area of the Runx2 (Figure 3G) and Bmp2 (Figure 3H) labeling in the BMSCs cultured on these scaffolds showed a gradually increasing trend, which indicated that both the MBG-NH_2_/IGF and MBG-NH_2_/IGF@SF/VEGF scaffolds could promote osteogenic protein expression in BMSCs, though the performance of the MBG-NH_2_/IGF@SF/VEGF scaffolds was better. Taken together, these results show that both the MBG-NH_2_/IGF and MBG-NH_2_/IGF@SF/VEGF scaffolds can increase the expression of osteoblast-specific marker proteins at the translation level, and the performance of the MBG-NH_2_/IGF@SF/VEGF scaffolds was superior to that of the MBG-NH_2_/IGF scaffolds.

### 2.4. The Effect of Scaffolds on Angiogenesis In Vitro

#### 2.4.1. Proliferation in HUVECs during Angiogenesis

To study the effect of the scaffolds on HUVECs during angiogenesis, we evaluated the effect of the scaffolds on HUVEC proliferation. Thus, we detected the bioactivity of HUVECs. The results showed that the MBG-NH_2_/IGF and MBG-NH_2_/IGF@SF/VEGF scaffolds had the best effects on HUVEC proliferation (Figure 4A).

#### 2.4.2. Expression of Angiogenic Genes

To determine the effect of these scaffolds on angiogenesis, we detected the mRNA expression levels of angiogenic-specific marker genes in HUVECs cultured on these scaffolds. The results showed that compared with those of the MBG-NH_2_ scaffolds, the mRNA levels of *ANG* (Figure 4B), *CD31* (Figure 4C), *HIF-1α* (Figure 4D), and *vWF* (Figure 4E) of the other two scaffolds were significantly higher in HUVECs, and the mRNA expression level of the HUVECs cultured on the MBG-NH_2_/IGF@SF/VEGF scaffolds was significantly higher than that on the MBG-NH_2_/IGF scaffolds. Thus, these results suggest that HUVECs angiogenesis could be promoted after culturing on the MBG-NH_2_/IGF@SF/VEGF scaffolds.

#### 2.4.3. Expression of Angiogenic Proteins

To further verify whether these scaffolds could promote angiogenesis at the protein expression level, we detected the expression of the CD31 protein in HUVECs cultured on these scaffolds with immunofluorescence. The result showed that compared with the other two scaffolds, the CD31 protein-labeled fluorescence intensity and area of the HUVECs cultured on the MBG-NH_2_/IGF@SF/VEGF scaffolds were stronger and significantly larger, respectively, and it could be seen that the thin filaments of the CD31 protein connected cells together and that the cells showed a similar behavior in the vascularization on the scaffolds (Figure 4F). Thus, these results suggest that the MBG-NH_2_/IGF@SF/VEGF scaffolds could enhance angiogenesis process in HUVECs after culturing on them.

### 2.5. Synergistic Effect of Osteogenesis and Angiogenesis

To study the effect of vascularization on osteogenesis, we co-cultured BMSCs and HUVECs with transwells, and we collected BMSC mRNA for RNA sequencing (RNA-seq). The results showed that the BMSCs co-cultured with HUVECs on the MBG-NH_2_/IGF@SF/VEGF scaffolds had a larger number of differentially expressed genes, accounting for 7.53% of the total gene, than those on the MBG-NH_2_ scaffolds. However, the proportion of differentially expressed genes in the BMSCs co-cultured with HUVECs on the MBG-NH_2_/IGF scaffolds to that on the MBG-NH_2_ scaffolds was only 1.33, and 5.58% of all annotated genes were differentially expressed in the BMSCs co-cultured on the MBG-NH_2_/IGF@SF/VEGF scaffolds compared with those on the MBG-NH_2_/IGF scaffolds (Figure 5A–D).

In addition, we used GSEA to detect the leading genes of the KEGG Notch and KEGG TGF-β signaling pathways of the BMSCs co-cultured with HUVECs. Compared with those on the MBG-NH_2_ scaffolds, the BMSCs grown on the MBG-NH_2_/IGF@SF/VEGF scaffolds showed a higher enrichment of both signaling pathways (Figure 5E,F). In addition, the BMSCs grown on the MBG-NH_2_/IGF@SF/VEGF scaffolds were more enriched in the KEGG MAPK (Figure 5G), KEGG Notch (Figure 5H), and KEGG TGF-β (Figure 5I) signaling pathways compared with the MBG-NH_2_/IGF scaffolds. The opposite signals of Wnt and TGF-β make osteogenesis and resorption reach equilibrium [46].

We found that 15 differentially expressed genes related to bone remodeling, bone development, bone morphogenesis, bone maturation and bone mineralization showed positive and negative regulations for the BMSCs co-cultured with HUVECs on the MBG-NH_2_/IGF@SF/VEGF scaffolds compared with those on the MBG-NH_2_ scaffolds (Figure 5J), and 19 differentially expressed genes showed positive and negative regulation for the BMSCs co-cultured with HUVECs on the MBG-NH_2_/IGF@SF/VEGF scaffolds compared with those on the MBG-NH_2_/IGF scaffolds (Figure 5M). In order to further verify the gene expression difference of the BMSCs co-cultured with HUVECs on the MBG-NH_2_/IGF@SF/VEGF, MBG-NH_2_/IGF and MBG-NH_2_ scaffolds, we selected two genes for RT-qPCR detection. It was shown that compared with that of the MBG-NH_2_ scaffolds, the mRNA expression level of *Csf1r* in the MBG-NH_2_/IGF@SF/VEGF scaffolds was significantly decreased and the mRNA expression level of *Grem1* was significantly increased (Figure 5K). Similarly, compared with the MBG-NH_2_/IGF scaffolds, the mRNA expression level of *Cthrc1* in the MBG-NH_2_/IGF@SF/VEGF scaffolds was significantly decreased and the expression level of *Cited2* was significantly increased (Figure 5N). Furthermore, we drew an interaction network between proteins’ differentially expressed genes and their osteogenic function compared with the BMSCs co-cultured with HUVECs on the MBG-NH_2_/IGF@SF/VEGF and MBG-NH_2_ scaffolds (Figure 5L). In conclusion, the MBG-NH_2_/IGF@SF/VEGF scaffolds can enhance the osteogenic process by promoting angiogenesis and, ultimately, bone defect repair.

### 2.6. In Vivo Evaluation of Bone Defect Repair Effects of Scaffolds

In order to systematically study the role of scaffolds in bone defect repair and angiogenesis, we constructed critical skull defect repair models in mice. Four weeks after the scaffolds were implanted into the skull defects of mice, the repair models of the bone defects were scanned with high-resolution micro-CT. The data demonstrated that compared with the control group without scaffolds, more new bone formed at the three bone defects filled with these scaffolds. Furthermore, the MBG-NH_2_/IGF and MBG-NH_2_/IGF@SF/VEGF scaffolds performed best on the bone defects (Figure 6A). The quantitative morphological analysis results further demonstrated that the amount of trabeculae (Tb. N) of the corresponding models of the MBG-NH_2_/IGF and MBG-NH_2_/IGF@SF/VEGF scaffolds was significantly higher than those of the MBG-NH_2_ scaffolds and the NC group (Figure 6B).

Furthermore, hematoxylin–eosin staining (HE) and Masson trichrome staining were used for the histochemical analysis of the bone defect repair model. The results of the HE staining showed that the bone defect edge of the MBG-NH_2_/IGF@SF/VEGF scaffold graft was connected with thicker soft tissue than the other two scaffold groups. In the NC group, there was little new bone formed at the edge of the defect and only a few thin soft tissue connections. After Masson’s trichrome staining, more collagen was formed on the bone defect edges of the MBG-NH_2_/IGF and MBG-NH_2_/IGF@SF/VEGF scaffolds, among which the MBG-NH_2_/IGF@SF/VEGF scaffolds performed better. (Figure 6C).

In order to further study the effect of the MBG-NH_2_/IGF@SF/VEGF scaffolds on angiogenesis and the effect of vascularization on osteogenesis in the bone defect repair model, Bmp2 and CD31 were stained in tissues. The results showed that Bmp2 was expressed in the bone defects filled with three scaffolds, and the amount of Bmp2 was highest in the MBG-NH_2_/IGF@SF/VEGF scaffold group. CD31 serves as a marker of angiogenesis. We found that there were more haversian canals in the bone defect grafted with the MBG-NH_2_/IGF@SF/VEGF scaffolds, as well as obvious CD31 expression (Figure 6D). In conclusion, the MBG-NH_2_/IGF@SF/VEGF scaffolds demonstrate good bone induction ability in bone defect repair and promote bone angiogenesis in the early stage of bone healing, thus providing a suitable microenvironment for subsequent osteogenesis.

## 3. Discussion

Bone defect repair is a complex physiological process. The process of neovascularization is also indispensable in osteogenesis. Our purpose was to construct a bone graft scaffold that could promote bone formation with angiogenesis. In this research, coaxial electrospinning was employed to prepare MBG-NH_2_/IGF@SF/VEGF scaffolds, with MBG-NH_2_ adsorbing IGF-1 as the core and SF adsorbing VEGF as the shell.

Previous studies have shown that due to its regular porosity and large specific surface area [19,20], MBG-NH_2_ has a strong biocompatibility, factor-binding capacity, and bone-promoting ability [23]. Moreover, SF has a good biocompatibility, high biodegradation rate, excellent mechanical properties, and the ability to promote the osteogenic differentiation of BMSCs, thus becoming a favorable scaffold material for bone defect repair. SF can also be combined with other biomaterials and processed into composite scaffolds, making it an excellent drug carrier [47,48]. In addition, the MBG-NH_2_/IGF@SF/VEGF scaffolds showed good properties in mechanical testing during our study, suggesting that they could withstand the pressure of bone and play a role in the repair of bone defects. (Figure 1D).

In a core–shell structure, it is favorable for SF and VEGF in the shell to promote angiogenesis first, transport nutrients for cells and carry away wastes during subsequent osteogenesis, and provide a good microenvironment for osteogenesis. MBG-NH_2_ can sustain the release of IGF-1 to further promote bone defect repair, prolong the time of scaffold bone repair, and not only repair the bone structure but also promote the recovery of the normal metabolic function of the bone. Both the phosphorylated mammalian target of rapamycin (p-mTOR) and MAPK signaling pathways could be activated to induce osteoblast differentiation [38,49,50,51]. Previous studies have shown that VEGF continuously induces the activation of Erk2 during differentiation but does not induce the activation of Erk1 [52,53,54], and phosphorylated Erk1/2 could phosphorylate the ser301 and ser319 residues of Runx2 to activate it [45]. IGF-1 can simultaneously induce the activation of Erk1/2 and mTOR. Moreover, compared with VEGF, IGF-1 may play a stronger role in promoting osteogenesis through the ERK1/2 and mTOR pathways [38,52,53,54]. In this study, the BMSCs cultured on the MBG-NH_2_/IGF@SF/VEGF scaffolds for 3 days showed significantly higher *Runx2* transcriptional activity (Figure 3E) than those cultured on other scaffolds, which was due to the large amount of VEGF released by the MBG-NH_2_/IGF@SF/VEGF scaffolds in the early stage. However, the BMSCs cultured on the MBG-NH_2_/IGF scaffolds for 5 days demonstrated the significantly highest expression of p-Erk1/2 and p-mTOR (Figure 3F) due to the slow-release effect of the IGF-1 adsorbed in MBG-NH_2_, which gradually increased the concentration of IGF-1 in the middle phase, thereby activating p-Erk1/2. Furthermore, the BMSCs co-cultured with HUVECs on the MBG-NH_2_/IGF@SF/VEGF scaffolds for 7 days demonstrated a higher signal enrichment of the KEGG MAPK pathway than those on the MBG-NH_2_/IGF scaffolds, according to RNA-seq (Figure 5G), due to the delayed release of IGF-1 by MBG-NH_2_ in the core of the MBG-NH_2_/IGF@SF/VEGF scaffolds at a later stage that interacted with VEGF to promote MAPK pathway activation; additionally, the process of angiogenesis could potentially activate the MAPK pathway to promote osteoblast differentiation. In addition, the expression of osteogenic marker genes of the BMSCs cultured on the MBG-NH_2_/IGF@SF/VEGF scaffolds for 7 days was significantly increased, including that of *Bmp2* (Figure 3A) and *Runx2* (Figure 3D), which were downstream of these pathways and had different but complementary roles in osteogenesis [55]. These results are consistent with the results of Bmp2 and Runx2 protein expression (Figure 3G,H). Furthermore, since IGF-1 plays an important role in coupling matrix biosynthesis to sustained mineralization and contributes to the acceleration of the process of bone repair [56] and the core–shell structure can further prolong the slow release process, the mineralization of the cells cultured on the MBG-NH_2_/IGF@SF/VEGF scaffolds for 14 days in vitro (Figure 1F) and in vivo (Figure 2C) also proved that the MBG-NH_2_/IGF@SF/VEGF scaffolds could continuously deliver growth factors to bone defects and had a more durable bone repair effect. These results are consistent with those of the in vivo experiments shown in Figure 6.

ANG plays an important role in promoting vascular growth and regeneration [57]. *CD31* mediates cell adhesion, which is conducive to the healing of wounds, while vWF can act as a mediator of platelets, adhere to collagen fibers, form thrombi, and stop bleeding [58]. Hif-1α is a central regulator of the adaptive response to hypoxia availability and is necessary for normal bone development [59], and inducing VEGF significantly enhances the angiogenic ability of HUVECs [39]. Here, HUVECs cultured on the MBG-NH_2_/IGF@SF/VEGF scaffolds showed high expression levels of the vascular marker genes mentioned above, and the mRNA expression level of *CD31* (Figure 4C) was consistent with its protein expression level (Figure 4F). CD31 is usually located at the tight junction between vascular endothelial cells and has a direct relationship with angiogenesis, and it is often used as a marker of angiogenesis [60]. In addition, the expression of CD31 in the bone defect grafted on the MBG-NH_2_/IGF@SF/VEGF scaffolds was also high in the haversian canal (Figure 6D), which might be neovascular. *Notch* expression not only promotes the proliferation and differentiation of osteoblasts [61] but is also an important positive regulator of bone vascular growth. Endothelial cells regulate *Notch* activity to promote bone angiogenesis combined with osteogenesis [62]. Our GSEA results showed that the KEGG Notch pathway signal enrichment of cells cultured on the MBG-NH_2_/IGF@SF/VEGF scaffolds was higher than that of the MBG-NH_2_ and MBG-NH_2_/IGF scaffolds. We conclude that the Notch pathway involved in the process of osteogenesis with angiogenesis treatment.

In this study, we prepared MBG-NH_2_/IGF@SF/VEGF scaffolds with coaxial electrospinning, and they were evaluated for osteogenic effect. The MBG-NH_2_/IGF@SF/VEGF scaffolds could promote mineralization both in vivo and *in vitro*. We simulated the transcriptional activity of Runx2 through the phosphorylated Erk1/2 activation pathway, and we detected the mRNA expression levels of osteogenic-specific marker genes and protein expression levels, which confirmed that the MBG-NH_2_/IGF@SF/VEGF scaffolds could promote osteogenesis. Additionally, the MBG-NH_2_/IGF@SF/VEGF scaffolds also demonstrated good performance in promoting angiogenesis. Furthermore, we found that the MBG-NH_2_/IGF@SF/VEGF scaffolds had a good repair effect on severe skull defects in mice in vivo, and VEGF and IGF-1 adsorption on scaffolds showed synergistic effects that led to greater ossification, larger trabeculae, and higher angiogenesis. In conclusion, these MBG-NH_2_/IGF@SF/VEGF scaffolds could promote angiogenesis and then promote bone defect repair, thus providing a new potential for bone tissue engineering.

## 4. Materials and Methods

### 4.1. Preparation of MBG-NH_2_ and SF

MBG was prepared using the sol–gel and template self-assembly methods [63,64]. Firstly, 9.6 g of P123 (PEO_20_-PPO_70_-PEO_20_, Macklin, Shanghai, China) was dissolved in H_2_O and stirred to obtain a clear solution, the pH of which was then adjusted to 1. Then, 16 g of 1,3,5-trimethylbenzene (TMB, Macklin, Shanghai, China) was added to the solution and mixed for 2 h. After that, 16 mL of tetraethyl orthosilicate (TEOS, Macklin, Shanghai, China), 7.5 g of Ca(NO_3_)_2_.4H_2_O, (Fengchuan, Tianjin, China), and 5 g of triethyl phosphate (TEP, Macklin, Shanghai, China) were added in sequence. The mixture was treated in a 45 °C water bath for 24 h, after which the PH was adjusted to 10. The mixture was allowed to react in a chemical reactor at 100 °C for 48 h. The collected precipitate, after centrifugation and washing, was calcined at 550 °C in muffle furnace for 6 h to acquire the MBG [18,65].

The MBG powder obtained in the previous step was mixed with 1 M aminopropyltriethoxysilane (APTES, Macklin, Shanghai, China). After reacting for 16 h, the mixture was washed with H_2_O and finally dried. The dried white powder was the MBG-NH_2_ [66].

Briefly, the SF solution was extracted from silkworm cocoon with a LiBr-saturated solution after removing sericin. Impurities were removed with dialysis and reverse dialysis using a 10% (*w*/*v*) polyethylene glycol (PEG 10000, Sangon Biotech, Shanghai, China) solution to obtain a pure solution with an appropriate concentration [15,67].

### 4.2. Preparation of Electrospun Scaffolds

MBG-NH_2_ powder was mixed with phosphate-buffered saline (PBS, pH = 7.4), and then IGF-1 (Peprotech, East Windsor, NJ, USA) was added before the mixture was incubated at 4 °C overnight. The mixture was then centrifuged to remove the supernatant to continue incubation in PBS for a gradient. The sustained release of IGF-1 was tested using an enzyme-linked immunosorbent assay kit (ELISA, Wuhan JYM Biological Technology, Wuhan, China), the sensitivity of which was less than 6.1 pg/mL [63]. The slow-release trend could be observed on a broken line graph drawn according to the concentration of IGF-1 measured at each time point. The concentration of IGF-1 was calculated with a standard curve according to the manufacturer’s instructions. Similarly, VEGF (Peprotech, East Windsor, NJ, USA) was added in the SF solution and then incubated at 4 °C for 12 h.

Poly (ε-caprolactone) (PCL, J&K Scientific, Beijing, China) was added to a 1,1,1,3,3,3-hexafluoro-2-propanol (HFIP, Macklin, Shanghai, China) solution, and a clarification solution (10%, *w*/*v*) was obtained after mixing for electrospin.

*In vivo* and in vitro research was carried out with the following three groups of samples:

Fiber made of MBG-NH_2_ (MBG-NH_2_ scaffolds);

Fiber made of MBG-NH_2_ adsorbed with IGF-1 (MBG-NH_2_/IGF scaffolds);

Fiber with MBG-NH_2_ adsorbed with IGF-1 as the core and SF adsorbed with VEGF as the outer layer (MBG-NH_2_/IGF@SF/VEGF scaffolds).

The MBG-NH_2_ scaffolds were simply prepared by electrospinning fibers with a mixture of 3 mg of MBG-NH_2_ powder and 3 mL of electrospinning solution. Then, 3 mL of the electrospinning solution was added to an MBG-NH_2_/IGF-1 or SF/VEGF solution, and uniform dispersion systems were obtained by stirring. Likewise, the MBG-NH_2_/IGF scaffolds were simply prepared by electrospinning fibers with a mixture of MBG-NH_2_/IGF-1 and electrospinning solution. Somewhat differently, the MBG-NH_2_/IGF@SF/VEGF scaffolds were synthesized with coaxial electrospinning with an MBG-NH_2_/IGF-1 mixture (core) and an SF/VEGF mixture (shell). Electrospinning was conducted with a 20 kV voltage. The electrospinning speed was generally kept at 0.75 mL/h and 0.5 mL/h for the coaxial fiber.

### 4.3. Morphology and Structure Characterization

The scaffolds were sprayed with gold for 30 s, and they were then observed with a scanning electron microscope (SEM; JEOL, JSM-7900F, Tokyo, Japan) in a vacuum.

Fluorescein-5-isothiocyannate (FITC, Origene, Rockville, MD, USA, 1:10) was used to mark the core and Alexa Fluor 594 (Origene, Rockville, MD, USA, 1:20) was used to mark the shell when preparing the scaffolds. The scaffolds were then observed with a confocal laser scanning microscope (CLSM, Leica TCS SP8, Bonn, Germany) [68].

The chemical structures and functional groups of the scaffolds were characterized by Fourier transform infrared (FTIR, Thermo Scientific, Waltham, MA, USA) spectra. The spectrum acquisition range was 400–4000 cm^−1^.

The tensile test was performed on the scaffolds with an ElectroForce (3220, Bose, New Castle, DE, USA), and the calculations of the linear slope of the stress–strain curve were used to obtain the elastic moduli of the scaffolds.

### 4.4. Mineralization Deposits in Simulated Body Fluid

*In vitro* mineralization was measured in an R-simulated body fluid (SBF) medium (142.0 mM Na^+^, 5.0 mM K^+^, 1.5 mM Mg^2+^, 2.5 mM Ca^2+^, 103.0 mM Cl^−^, 27.0 mM HCO_3_^−^, 1.0 mM HPO4^2−^, and 0.5 mM SO4^2−^) [69]. The MBG-NH_2_, MBG-NH_2_/IGF and MBG-NH_2_/IGF@SF/VEGF scaffolds were separately immersed in the SBF medium and incubated at 37 °C for 14 days. The surfaces of the mineralization deposits on three groups of scaffolds were characterized with SEM, which could indicate the formed mineralization deposits.

### 4.5. Cells and Cell Culture

#### 4.5.1. BMSCs

BMSCs (BeNa Culture Collection, Kunshan, China) were cultured with a high-glucose Dulbecco’s modified eagle medium (DMEM, Cytiva, Marlborough, MA, USA) supplemented with 10% fetal bovine serum (FBS, Sijiqing, Beijing, China), 100 units/mL of penicillin, and 100 μg/mL of streptomycin. The mineralization-inducing medium contained DMEM with 50 μg/mL of ascorbic acid (VC), 10 nM dexamethasone (DEX), and 10 mM sodium β-glycerophosphate (β-Gly) [69]. BMSCs were incubated under a 5% CO_2_ atmosphere at 37 °C, with the culture medium replaced every three days.

#### 4.5.2. HUVECs

HUVECs were cultured with a 1640 basic RPMI medium (Gibco, Grand Island, NY, USA) supplemented with 10% FBS and the same antibiotics as BMSCs. HUVECs were incubated under a 5% CO_2_ atmosphere at 37 °C, with the culture medium replaced every two days.

### 4.6. Cell Proliferation

BMSCs were seeded at a density of 2.5 × 10^4^ cells/mL in 96-well plates on the MBG-NH_2_, MBG-NH_2_/IGF and MBG-NH_2_/IGF@SF/VEGF scaffolds. The three groups of cells were together incubated for different days, with five individual wells for every group corresponding to specific culture days. The proliferation was measured under adsorbance at 450 nm by adding a Cell Counting Kit-8 (CCK-8, Beyotime Biotech, Haimen, China) to the cell medium, followed by 4 h of incubation at 37 °C [70].

### 4.7. Cell Differentiation

As for cell proliferation, BMSCs were seeded at a density of 2.5 × 10^4^ cells/mL in 96-well plates on the MBG-NH_2_, MBG-NH_2_/IGF and MBG-NH_2_/IGF@SF/VEGF scaffolds. The three groups of cells were together incubated for different days. What differed was that two sets of duplicate samples were required for proliferation and differentiation each. The cell differentiation rate was then determined by alkaline phosphatase (ALP) activity. An ALP working fluid, which can hydrolyze colorless *p*-nitrophenol inorganic phosphate (PNPP) to yellow *p*-nitrophenol (PNP), was prepared by mixing a 0.1 M NaHCO_3_-Na_2_CO_3_ buffer (pH = 10.0), 0.1% Triton X-100, 2 mM MgCl_2_, and 6 mM PNPP. Then, a 100 μL working solution was added to a single well 24 h in advance. We calculated the value of absorbance at 405 nm divided by the value of absorbance at 450 nm (A_405_/A_450_) to determine the relative ALP activity [18].

### 4.8. Cell Mineralization

The cell mineralization capacity was assessed via the detection of calcium deposits. BMSCs were plated on the MBG-NH_2_, MBG-NH_2_/IGF and MBG-NH_2_/IGF@SF/VEGF scaffolds at a density of 2.5 × 10^4^ cells/mL in 6-well plates. After 3 days of normal culture, the cells were transferred to the mineralization-inducing medium to culture. After fixed with 70% ethanol for 10 min, an alizarin red solution (40 mM, pH = 4.2) was used to stain BMSCs at 37 °C for 15 min. Finally, after several washings, images of calcium deposits were obtained through stereoscope and microscopes.

### 4.9. Transfection and Luciferase Assay

With the luciferase assay, the Runx2 transcriptional activity was detected through Runx2 binding element-Luc plasmid. The plasmid was transfected by Lipofectamine 2000 (Invitrogen, Waltham, MA, USA) into BMSCs cultured on the MBG-NH_2_, MBG-NH_2_/IGF and MBG-NH_2_/IGF@SF/VEGF. A Dual Luciferase Assay Kit (Beyotime Biotech, Haimen, China) was used to measure luciferase activities 3 days later through Fluoroskan Ascent FL (Thermo scientific, Waltham, MA, USA), and the relative luciferase activities were calculated against an internal control.

### 4.10. Quantitative Real-Time Polymerase Chain Reaction (RT-qPCR)

A Trizol (Invitrogen, USA) reagent was applied to extract the total mRNA from the BMSCs and HUVECs cultured on the MBG-NH_2_, MBG-NH_2_/IGF and MBG-NH_2_/IGF@SF/VEGF scaffolds for 7 days. Afterwards, cDNA was synthesized by SuperScript III (Waltham, MA, USA). RT-qPCR was reacted with cDNA through SYBR Green (Applied Biosystems, Waltham, MA, USA). The forward and reverse primers were used as follows: bone morphogenetic protein 2 (*Bmp2*) 5′-CTGACCACCTGAACTCCAC-3′ and 5′-CATCTAGGTACAACATGGAG-3′; osteopontin (*Opn*) 5′-TCCAAAGCCAGCCTGGAAC-3′ and 5′-TGACCTCAGAAGATGAACTC-3′; *Osterix* 5′-GTCAAGAGTCTTAGCCAAACTC-3′ and 5′-AAATGATGTGAGGCCAGATGG-3′; runt-related transcription factor 2 (*Runx2*) 5′-GAATGCACTACCCAGCCAC-3′ and 5′-TGGCAGGTACGTGTGGTAG-3′; glyceraldehyde 3-phosphate dehydrogenase (*GAPDH*) 5′-CATGGCCTTCCGTGTTCCTA-3′ and 5′-CCTGCTTCACCACCTTCTTGAT-3′ [70]; angiotensin (*ANG*) 5′-GTGCTGGGTCTGGGTCTGAC-3′ and 5′-GGCCTTGATGCTGCGCTTG-3′; platelet endothelial cell adhesion molecule-1 (*CD31*) 5′-CAACGAGAAAATGTCAGA-3′ and 5′-GGAGCCTTCCGTTCTAGAGT-3′; hypoxia-inducible factor-1α (*HIF-1a*) 5′-CCAGATCTCGGCGAAGTAAAG-3′ and 5′-GCTGATGGTAAGCCTCATCAC-3′; Von Willebrand factor (*vWF*) 5′-CCCCTGAAGCCCCTCCTCCTA-3′ and 5′-ACGAACGCCACATCCAGAACC-3′; Colony-Stimulating Factor 1 Receptor (*Csf1r*) 5′-CCTGCGATGTGTGAGCAATG-3′ and 5′-CGGATAATGAACCCTCGCCA-3′; Gremlin 1 (*Grem1*) 5′-GAATCGCACCGCATACACTG-3′ and 5′-TGGCTCCTTGGGAACCTTTC-3′; Collagen Triple Helix Repeat Containing Protein 1 (*Cthrc1*) 5′-GGTCGGGATGGATTCAAAGG-3′ and 5′-AGCGAACTCCACGAACACTG-3′; and Cbp/p300-interacting transactivator, with Glu/Asp-rich carboxy-terminal domain 2 (*Cited2*) 5′-CTCATGGGCGAGCACATACA-3′ and 5′-GAGTTGTTAAACCTGGCGGC-3′. The whole process of RT-qPCR included the following steps: first 95 °C for 10 min, 56 °C for 60 s, and 72 °C for 30 s; followed 44 cycles of 95 °C for 30 s, 56 °C for 60 s, and 72 °C for 30 s; finally, 4 °C to stop.

### 4.11. Western Blot

BMSCs were planted on the MBG-NH_2_, MBG-NH_2_/IGF and MBG-NH_2_/IGF@SF/VEGF scaffolds in a 6-well plate at a density of 1 × 10^5^ cells/mL. After 5 days of culture, the collected cells were washed with PBS and then lysed (lysate: EDTA: phosphatase inhibitor: protease inhibitor = 50:1:1:1) on ice. The protein was obtained after centrifuging at 12,000× *g* for 15 min at 4 °C. After determining the protein concentration with a BCA protein assay kit (Thermo, Waltham, MA, USA), 30 μg of protein was mixed with a ×6 loading buffer and lysate before being heated in a water bath to denature it. Then, 10% sodium dodecyl sulfate (SDS)–polyacrylamide gel electrophoresis was performed on the denatured protein, and the protein bands were transferred to a polyvinylidene fluoride (PVDF) membrane. The bands were stained with reed red and blocked with 5% (*w*/*v*) skim milk. After incubation for 2 h with Erk1/2 (Beyotime Biotech, Haimen, China, 1:1000), phospho-Erk1/2 (p-Erk, Cell Signaling Technology, Danvers, MA, USA, 1:2000), mTOR (Abcam, Cambridge, UK, 1:2000), and phospho-mTOR (p-mTOR, Abcam, Cambridge, UK, 1:2000), the PVDF membrane was treated with the corresponding species source of the second antibody at a 1:5000 dilution. Immunoreactivity was determined with enhanced chemiluminescent substrate (Pierce, MO, USA) for HRP detection.

### 4.12. Cell Immunofluorescence

The scaffolds were laid on the coverslips and placed in the 6-well plate. BMSCs and HUVECs were planted on the MBG-NH_2_, MBG-NH_2_/IGF and MBG-NH_2_/IGF@SF/VEGF scaffolds at a density of 5 × 10^4^ cells/mL and cultured for 7 days. The washed cells were fixed with acetone and methanol (1:1). Then, 10% (*v*/*v*) horse serum was used for blocking. After being treated with the primary antibody against Bmp2 (Beyotime Biotech, Haimen, China, 1:200), Runx2 (Cell Signaling Technology, USA, 1:1600), CD31(Cell Signaling Technology, Danvers, MA, USA, 1:200) and β-actin (Beyotime Biotech, Haimen, China, 1:20) overnight, β-actin was labeled with FITC (1:100), while Bmp2, Runx2 and CD31 were labeled with Alexa Fluor 594 (1:100). Then 4′,6-diamidino-2-phenylindole (DAPI, Beyotime Biotech, Haimen, China) was used to stain the nucleus. Finally, 90% glycerol was used for sealing, and the sample was observed under CLSM.

### 4.13. Gene Sequencing

BMSCs and HUVECs were co-cultured through transwells on the MBG-NH_2_, MBG-NH_2_/IGF and MBG-NH_2_/IGF@SF/VEGF scaffolds, and the total extracted mRNA was sequenced to study the effect of osteogenesis under the condition of vascular growth. HUVECs were seeded in 6-well plates with a density of 1 × 10^5^ cells/mL and incubated for 6 days. BMSCs were seeded in transwells at the same density. After cells adhered, transwells were transferred to the 6-well plate so that the BMSCs and HUVECs could be co-cultured for 7 days. Finally, the mRNA of cells was separately collected with Trizol, and the mRNA of the BMSCs was then sequenced on the RNA-Seq platform (NovaSeq 6000, Illumina, San Diego, CA, USA) and analyzed via hierarchical clustering, volcano plots, gene set enrichment analysis (GSEA), and gene interaction.

### 4.14. In Vivo Studies

Critical-size cranial defects were induced in C57/BL6 mice (male) aged 6 weeks. Five mice as a group were used for one kind of scaffold transplantation. Holes with a diameter of 2.35 mm were drilled with a dental drill (1500 rpm) on both sides of the mouse skull. The MBG-NH_2_, MBG-NH_2_/IGF and MBG-NH2/IGF@SF/VEGF scaffolds were implanted into the left hole, and the right hole was inserted with nothing as a negative control. Mice skulls were irradiated with microcomputed tomography (micro-CT) after 4 weeks to obtain the trabecular number (Tb.N). Then, the mice skulls were fixed with 4% paraformaldehyde and decalcified in a 10% EDTA solution for 1 month. Afterwards, skulls were dehydrated with a gradient alcohol and embedded in paraffin to make them into 4 μm thick sections that were stained with hematoxylin and eosin (HE), Masson’s trichrome, Bmp2 (Affinity, San Francisco, CA, USA, 1:200), and CD31 (Affinity, San Francisco, CA, USA, 1:100) before finally being observed under a microscope (NanoZoomer S360, Hamamatsu, Japan).

### 4.15. Statistics Analysis

All the statistics in the figures are reported as mean ±SE derived from three to five independent experiments. Statistical significance, which was calculated with the two-sample t-test, was valid when *p* < 0.05.

## Data Availability

The data presented in this study are available on request from the corresponding author.

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
