# Peer review of "The Effect of Angiogenesis-Based Scaffold of MesoporousBioactive Glass Nanofiber on Osteogenesis"

_ijms, 2022, doi:10.3390/ijms232012670_

Round 1

Reviewer 1 Report

Dear Authors, the comments and the questions ralated to the manuscript are provided in supplemented PDF file.

Reviewer 2 Report

The article “The effect of angiogenesis-based scaffold of Mesoporous bioactive glass nanofiber on osteogenesis” is devoted to development of novel bone scaffold that may heal critical-sized bone defects accelerating osteo- and angiogenesis. There are some issues that need to be considered:

Line 153 “To study the loading capacity of MBG-NH2, we tested the release ability for IGF-1.” Loading capacity and release are different parameters, and may not be connected directly (e.g., if loaded drug binds covalently to the material, or reveals strong adsorption forces). Loading capacity (LC%) is usually calculated by the amount of total entrapped drug divided by the total material weight.

It is suggested to provide a table with studied samples, their abbreviation and characterisation. This will improve readability.

Figure 1 C, D, E: the graphs are not readable; the fonts and scale should be increased.

Figure 4 F: scale bar is not readable.

For all Figures: the font size should be the same within the Figure.
